# ODG: Occupancy Prediction Using Dual Gaussians

**Yunxiao Shi**[†]    **Yinhao Zhu**[†]    **Shizhong Han**[†]    **Jisoo Jeong**[†]    **Amin Ansari**[‡]
**Hong Cai**[†]    **Fatih Porikli**[†]

[†]Qualcomm AI Research[*]   [‡]Qualcomm Technologies, Inc

{yunxshi,yinhaoz,shizhan,jisojeon,amina,hongcai,fporikli}@qti.qualcomm.com

## Abstract

Occupancy prediction infers fine-grained 3D geometry and semantics from camera images of the surrounding environment, making it a critical perception task for autonomous driving. Existing methods either adopt dense grids as scene representation which is difficult to scale to high resolution, or learn the entire scene using a single set of sparse queries, which is insufficient to handle the various object characteristics. In this paper, we present ODG, a hierarchical dual sparse Gaussian representation to effectively capture complex scene dynamics. Building upon the observation that driving scenes can be universally decomposed into static and dynamic counterparts, we define dual Gaussian queries to better model the diverse scene objects. We utilize a hierarchical Gaussian transformer to predict the occupied voxel centers and semantic classes along with the Gaussian parameters. Leveraging the real-time rendering capability of 3D Gaussian Splatting, we also impose rendering supervision with available depth and semantic map annotations injecting pixel-level alignment to boost occupancy learning. Extensive experiments on the Occ3D-nuScenes and Occ3D-Waymo benchmarks demonstrate our proposed method sets new state-of-the-art results while maintaining low inference cost.

## 1 Introduction

3D spatial understanding forms the foundation of autonomous systems such as self-driving cars. 3D object detection [57, 23, 34, 33] has been the primary task that outputs bounding boxes to capture different entities in the scene. Concise as box representation is, it cannot deal with out-of-vocabulary or irregularly-shaped objects (*e.g.* trash can on the side of road, excavator with arms deployed) which is critical for driving safety. Calling for a unified 3D representation that can handle such cases, 3D occupancy adopts a voxel grid to partition the scene and jointly predicts the occupancy state and semantic labels of each voxel, which provides critical information for downstream planning [22].

Previous approaches have explored regular grids like voxel [58, 63], BEV ground plane [31, 61], and tri-perspective plane [25] to represent the scene followed by dense classification. Such approaches do not take into account the fact that most of the voxels are empty [3, 50] and allocate equal resource for each one, which inevitably results in severe inefficiency. To overcome such drawbacks, another line of research [47, 53] formulate the task of 3D occupancy as direct set prediction, effectively predicting 3D occupancy as set of sparse points from sparse latent vectors. Such sparse representation avoids spending resource to model empty regions and improves scalability. Recent works [27, 4, 10] utilize 3D Gaussians instead of points which have more spatial context, and also leverage the real-time rendering of 3D Gaussian Splatting (3DGS) [28, 29] to either complement learning from occupancy ground-truth or instead directly learn occupancy from 2D labels [59, 21] without relying on 3D annotations. Such sparse query-based methods have shown great promises for occupancy prediction.

---

[*]Qualcomm AI Research is an initiative of Qualcomm Technologies, Inc.

*But Is it really sensible to rely on a single set of queries to predict everything within a driving scene?* We observe that driving scenes can be universally decomposed into static background (*e.g.* roads, buildings, etc.) and dynamic agents (*e.g.* vehicles, pedestrians, etc.), and argue that the dynamic agents should have resources dedicated to them given their importance in occupancy prediction. Hence, we introduce a dual Gaussian query design, consisting of two sets of Gaussian queries, to model the static and dynamic parts of the scene for better capturing complex scene dynamics. To establish communication between queries, we propose a simple and effective attention scheme to achieve this. Meanwhile for Gaussian-based representation, it is critical to have a sufficient number of Gaussians in order to have enough capacity to represent the scene. But existing methods [27, 4] utilize a single transformer which can only handle a smaller number of Gaussians. Therefore we propose to predict Gaussians in a coarse-to-fine manner with a series of transformer layers, enabling the use of a much larger number of Gaussians and thereby increasing model capacity. In addition to learning from 3D occupancy ground-truth, we leverage the efficient rendering capabilities of 3DGS to generate the depth and semantic maps for each camera view at every stage, which are supervised by the corresponding 2D labels and improves model consistency. Our contributions can be summarized as follows:

- **Dual Gaussian Query Design:** We propose a novel dual-query architecture comprising two distinct sets of Gaussian queries to separately model the static and dynamic parts of the scene. A cross query attention is also introduced to establish effective interaction between queries, enhancing 3D occupancy prediction.

- **Hierarchical Coarse-to-Fine Refinement:** We refine the Gaussian properties in a hierarchical coarse-to-fine fashion, allowing a much larger number of 3D Gaussians to be utilized, effectively increasing model capacity and expressiveness.

- **Multi-Stage Rendering Supervision:** We enforce depth and semantic consistency through multi-stage real-time rendering using 3D Gaussian Splatting. This allows supervision from 2D labels across views, improving spatial coherence and prediction accuracy.

- **State-of-the-Art Performance:** Extensive experiments on the Occ3D [50] benchmark demonstrates that our proposed method sets new state-of-the-art results in 3D occupancy prediction, while maintaining competitive inference runtime.

## 2 Related Works

### 2.1 3D Occupancy Prediction

3D occupancy prediction partitions the 3D space with a voxel grid and simultaneously estimates the occupancy state and semantic label of each voxel. Earlier methods [7, 31, 25, 58, 63] utilize dense grids (*e.g.* voxel, BEV) as scene representation and learns 3D occupancy from ground-truth data. Given the high cost [56] of curating occupancy annotations, [62, 41, 5, 44] advocates the idea of using 2D labels projected from LiDAR or generated by vision foundation models [59, 21] to train 3D occupancy networks eliminating reliance on direct 3D occupancy annotations. Several works [8, 62, 24, 36] also explore self-supervised learning based on the photometric consistency between neighboring frames to learn 3D occupancy. Meanwhile, multiple 3D occupancy benchmarks [3, 58, 50, 51, 13, 56, 64] have been created based on existing datasets [17, 16, 6, 45].

### 2.2 Set Prediction with Transformers

Direct set prediction with transformers [52, 14] for perception tasks was first introduced in DETR [9]. Follow-up works like [57, 37, 34, 33] further advanced this technique for 3D object detection and obtained impressive results. Witnessing such success, [47, 53] adapted such set prediction paradigm for 3D occupancy prediction and demonstrated a competitive alternative to the common pipeline of explicit space modeling (*e.g.*, dense grids) followed by classification. We adopt such paradigm but different from previous works [34, 47, 53] that only use one set of queries assuming scene homogeneity, we define two distinctive sets of queries to better handle the dynamic agents in the scene.

### 2.3 Neural Rendering and 3D Gaussian Splatting

Neural rendering [48] aims to learn 3D representations from 2D data and experienced significant growth over the past few years [49]. Neural Radiance Fields (NeRF) [40, 1, 2] is one such technique

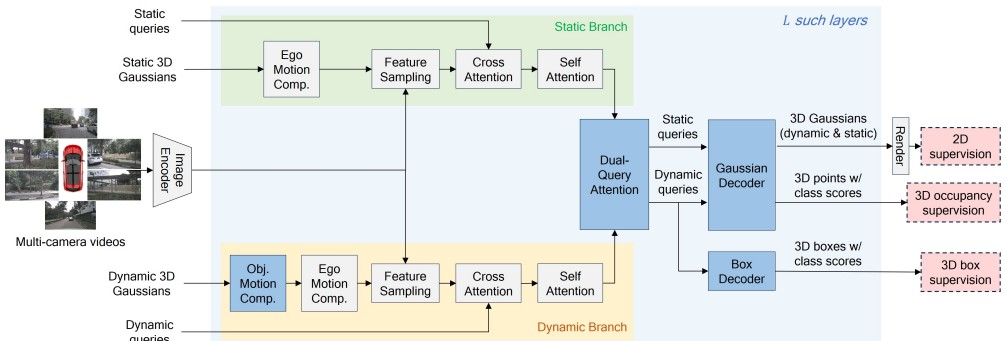

Figure 1: Overview of proposed ODG, where we model the dynamic and static elements of the scene with two separate sets of Gaussian queries. Dual-query attention aggregates information across dynamic and static queries. These queries are then decoded into 3D Gaussians, 3D points, as well as 3D bounding boxes (from dynamic queries only), which are supervised by ground-truth depth and semantic maps through rendering, ground-truth 3D occupancy, and ground-truth bounding boxes, respectively.

and has achieved impressive results. However, NeRF-based methods suffer from slow training and high memory cost. Recently, 3D Gaussian Splatting (3DGS) [28, 29] emerged as a rasterization pipeline which drastically reduced rendering time while also achieving incredible rendering quality. Earlier methods [60, 19] focused on improving per-scene rendering quality. Later generalizable 3DGS [11, 12, 46] were proposed to enable reconstructing large-scale scenes in a feed-forward manner. Given that 3D occupancy also aims to reconstruct the scene, latest research begin exploring 3D Gaussians for occupancy prediction [26, 27, 10, 4]. GaussianFormer [26] and GaussRender [10] voxelizes 3D Gaussians which incurs high compute cost. GaussTR [27] and GaussianFlowOcc [4] employs a single transformer to predict Gaussian parameters from sparse queries, which can only handle small number of Gaussians limiting model capability. In contrast, our method predicts Gaussians in a hierarchical coarse-to-fine fashion allowing a much larger number of Gaussians, effectively resulting in higher learning capacity.

## 3 Method

In this section, we present our proposed 3D occupancy approach, ODG, in which we adopt a dual Gaussian query design to capture the respective dynamic and static elements in the scene, as discussed in Sec. 3.2. We model object motions for the dynamic Gaussians (Sec. 3.3) and leverage attention to enable feature interaction between the dual queries (Sec. 3.4). Finally, we describe the training objectives in Sec. 3.5. Fig. 1 provides an overview of ODG. Next, we first provide a quick recap on the problem of 3D occupancy prediction.

### 3.1 Problem Definition

Given an ego-vehicle at time $T$, the task of 3D occupancy prediction takes $N_c$ multi-camera images (with $k \times N_c$ optional history frames where $k \geq 0$), $\mathbf{I} = \{I_c^t\}_{t=T-k,c=1}^{T,N_c}$ and corresponding camera parameters as input, and predicts a 3D semantic voxel grid $\mathbf{O} = \{c_1, c_2, \ldots, c_C\}^{H \times W \times Z}$ where $H, W, Z$ denotes the grid resolution and $C$ is the number of classes. Formally, 3D occupancy prediction can be defined as

$$\mathbf{O} = G(\mathbf{V}), \quad \mathbf{V} = F(\mathbf{I}), \tag{1}$$

where $F(\cdot)$ consists of an image backbone that extract multi-camera features and transforms them into scene representation $\mathbf{V}$. $G(\cdot)$ is another neural network that maps $\mathbf{V}$ to final occupancy predictions. Dense grids (*e.g.* voxel grid [58, 63], BEV/TPV grid [31, 25]) are common choices for $\mathbf{V}$ which are difficult to scale and cannot differentiate different object scales. Inspired by successes in object detection [9, 57], more recent works [35, 53] cast 3D occupancy prediction as direct set prediction with transformers, without explicitly building $\mathbf{V}$. At its core, a set of learnable queries $\mathbf{Q}$ and 3D points $\mathbf{P}$ are initialized to regress point locations and corresponding semantic class scores $\mathbf{C}$ simultaneously. Such a paradigm can be described as

$$\min_{\mathbf{P}, \mathbf{C}} D_p(\mathbf{P}, \mathbf{P}_g) + D_c(\mathbf{C}, \mathbf{C}_g), \tag{2}$$

where $\{\mathbf{P}_g, \mathbf{C}_g\}$ is the constructed ground-truth set for occupied voxels with $|P_g| = |C_g| = V_g$ being the number of occupied voxels. Each $p_g \in \mathbf{P}_g$ represents the voxel center coordinate and $c_g \in \mathbf{C}_g$ the class label. Correspondingly, $\{\mathbf{P}, \mathbf{C}\}$ denotes predictions where each $p_i \in \mathbf{P}$ and $c_i \in \mathbf{C}$ is predicted by query $q_i \in \mathbf{Q}$. $D_p(\cdot)$ and $D_c(\cdot)$ are geometric and semantic distances respectively.

### 3.2 Dual Dynamic and Static Gaussian Queries

Instead of predicting occupancy as point sets from one set of learnable queries as in Eq. 2, we define dual Gaussian queries as shown in Fig. 1 which are detailed below.

Formally, a standard 3D Gaussian $\mathbf{g}$ is parameterized by a set of properties

$$\mathbf{g} = \{\boldsymbol{\mu}, \mathbf{s}, \mathbf{r}, \sigma\}, \tag{3}$$

where $\boldsymbol{\mu} \in \mathbb{R}^3$ is the mean, $\mathbf{s} \in \mathbb{R}^3$ is the scale, $\mathbf{r} \in \mathbb{R}^4$ is the quaternion and $\sigma \in [0, 1]$ is the opacity. We define two sets of Gaussian queries for static and dynamic objects, each query contains a set of Gaussians and a feature vector

$$\mathbf{G}^s = \{\mathbf{g}^s_{i,k}\}_{i=1,k=1}^{S,K_\ell}, \mathbf{Q}^s = \{\mathbf{q}^s_i\}_{i=1}^{S}, \mathbf{q}^s_i \in \mathbb{R}^M; \mathbf{G}^d = \{\mathbf{g}^d_{j,k}\}_{j=1,k=1}^{D,K_\ell}, \mathbf{Q}^d = \{\mathbf{q}^d_j\}_{j=1}^{D}, \mathbf{q}^d_j \in \mathbb{R}^N, \tag{4}$$

where $S, D$ is the number of static and dynamic Gaussian queries, and $N, M$ is the embedding dimension of query features, $K_\ell$ is the number of Gaussians per query in stage $\ell$. Here we set $N = M$ for simplicity but they can be different.

For a dynamic query $\mathbf{g}^d$, whose intended purpose is to model the dynamic agents in the scene, what additional information should they have on top of Eq. 3 that can effectively accomplish this? We observe that the box representation from 3D object detection [57, 34, 33] is a good candidate that is tailored to capture dynamic objects. Therefore we expand the standard definition in Eq. 3 and define our dynamic Gaussian queries $\mathbf{g}^d$ as

$$\mathbf{g}^d = \mathbf{g}^s \oplus \mathbf{b}, \quad \mathbf{b} = [l, w, h, \theta, v_x, v_y, v_z], \tag{5}$$

where $l, w, h$ are the spatial dimensions of the 3D box and $\theta$ its rotation. $v_x, v_y, v_z$ is the velocity vector. Given for driving scenes, motion along the $z$-axis is negligible hence we set $v_z = 0$ and treat it as constant.

We initialize Gaussian means $\mathbf{g}^s_{:\mu}$ and $\mathbf{g}^d_{:\mu}$ from $\mathcal{U}[0, 1]$ for both types of queries and the box attributes $\mathbf{b}$ for $\mathbf{g}^d$ but leave the rest of the Gaussian properties uninitialized. Instead we predict them using separate MLPs with their means and query features

$$\{\mathbf{s}^d, \mathbf{r}^d, \sigma^d\} = \Phi(\mathbf{G}^d_{:\mu}, \mathbf{Q}^d), \{\mathbf{s}^s, \mathbf{r}^s, \sigma^s\} = \Phi(\mathbf{G}^s_{:\mu}, \mathbf{Q}^s), \tag{6}$$

where $\Phi$ denotes respective MLPs. To enable an sufficient number of Gaussians, unlike previous methods [27, 4] which only utilizes a single transformer that maintains the same number of points over layers, we predict Gaussians in a hierarchical coarse-to-fine fashion through a series of transformer-based layers $\mathcal{T}$ akin to [34, 53]

$$\mathbf{G}^{d,b}_\ell, \mathbf{C}^{d,b}_\ell, \mathbf{G}^d_\ell, \mathbf{C}^d_\ell, \mathbf{Q}^d_\ell = \mathcal{T}_\ell(\mathbf{G}^d_{:\mu\oplus\mathbf{b},\ell-1}, \mathbf{Q}^d_{\ell-1}; \Phi_s, \Phi_r, \Phi_\sigma), \tag{7}$$

$$\mathbf{G}^s_\ell, \mathbf{Q}^s_\ell, \mathbf{C}^s_\ell = \mathcal{T}_\ell(\mathbf{G}^s_{:\mu,\ell-1}, \mathbf{Q}^s_{\ell-1}; \Phi_s, \Phi_r, \Phi_\sigma), \tag{8}$$

where $1 \leq \ell \leq L$ with $L$ being the number of layers used. For each layer $\mathcal{T}_\ell$, it takes as input static Gaussian means $\mathbf{G}^s_{:\mu,\ell-1}$ and query features $\mathbf{Q}^s_{\ell-1}$ from the previous layer, and predict the current static Gaussian means $\mathbf{G}^s_{:\mu,\ell} \in \mathbb{R}^{S \times K_\ell \times 3}$ and corresponding class scores $\mathbf{C}^s_\ell \in \mathbb{R}^{S \times K_\ell \times C}$, where $K_{\ell-1} < K_\ell$ is the number of Gaussians per query at each stage which enables coarse-to-fine prediction. For dynamic Gaussians, besides decoding $\mathbf{G}^d_\ell \in \mathbb{R}^{D \times K_\ell \times 3}$ and $\mathbf{C}^d_\ell \in \mathbb{R}^{D \times K_\ell \times C}$, we also parallelly predict box attributes and class scores $\mathbf{G}^{d,b}_\ell \in \mathbb{R}^{D \times 10}$ and $\mathbf{C}^{d,b}_\ell \in \mathbb{R}^{D \times C}$ as shown in Eq. 7. We note that we do not perform coarse-to-fine prediction for boxes given the relatively small number of boxes that are typically present in the scene. The rest of the Gaussian properties is predicted by separate MLPs $\Phi$ as defined in Eqs. 6.

We aim to reduce the spatial artifacts of 3D occupancy through projective constraints. To this end, we render depth and semantic maps of each camera view at current time (keyframe) which effectively

enforces geometric and semantic consistency. We leverage 3D Gaussian splatting [28, 29] which offers real-time rendering:

$$\hat{D}_p = \sum_{i=1}^{G} T_i \sigma_i \mathbf{d}_i, \quad \hat{S}_p = \sum_{i=1}^{G} T_i \alpha_i \mathbf{c}_i, \tag{9}$$

where $p$ indicates a certain pixel location and $G = D + S$ is the number of Gaussians. $\sigma_i$ is the opacity and $T_i$ is accumulated transmittence. $\mathbf{c}_i \in [0, 1]^C$ is the "semantic probability" of the $i$-th Gaussian and $\mathbf{d}_i$ is the depth of the $i$-th Gaussian. $\hat{D}_p, \hat{S}_p$ are the rendered depth and semantic values at location $p$. We refer readers to [28] for further details regarding 3D Gaussian splatting.

### 3.3 Motion Modeling of Dynamic Gaussians

For each Gaussian $\mathbf{g}$ with $\mathbf{g}_{:\mu}$ representing its position in 3D space, it first samples 3D points following [34, 53]. Then we sample image features by projecting sampled 3D points onto each image feature plane using available camera extrinsics and intrinsics, and aggregate corresponding image features. Under the setting of having history frames, it is critical to move the Gaussians according to its motion to sample features correctly. For driving scenes we consider two types of motions: 1) ego-motion which is the motion of the ego-vehicle as it navigates across the scene, which effectively is camera motion, and 2) object-motion which describes how the dynamic agents themselves move in the scene.

For static Gaussians, compensating ego-motion is enough. For the motion of dynamic Gaussians, we adopt the approach of approximating instantaneous velocity with average velocity in a short-time window as in [34], and warp sampling points to previous timestamps $t_{-i}$ using the velocity vector $[v_x, v_y]$ from the dynamic Gaussian queries (Eq. 5)

$$x_{-i} = x_0 - v_x \cdot (t_0 - t_{-i}), \tag{10}$$
$$y_{-i} = y_0 - v_y \cdot (t_0 - t_{-i}), \tag{11}$$

where $t_0$ is the current timestamp. We note that on the $z$-axis, we assume there is no velocity give the nature of driving. Then we warp each dynamic Gaussian using camera motion

$$(x'_{-i}, y'_{-i}, z'_{-i}, 1)^\top = P_{-i}^{-1} P_0 (x_{-i}, y_{-i}, z_{-i}, 1)^\top,$$

where $P_{-i}, P_0$ are the ego poses at timestamp $t_{-i}$ and $t_0$.

### 3.4 Attention across Dynamic and Static Queries

To enable effective interaction between dynamic Gaussian queries $\mathbf{Q}^d$ and static Gaussian queries $\mathbf{Q}^s$, we first concatenate their features representations. We then apply Self-Attention [52] to the combined features, allowing for rich information exchange cross both query types. We refer to this mechanism as Dynamic-and-Static (DaS) Attention. This approach not only facilitates self-attention within the dynamic and static queries individually but also enables bidirectional cross-attention between them, enhancing the integration of dynamic and static information.

$$\mathbf{Q} = \text{Self-Attention}(\text{Concatenate}(\mathbf{Q}^d, \mathbf{Q}^s)), \tag{12}$$
$$\mathbf{Q}^d = \mathbf{Q}_{:D}, \ \mathbf{Q}^s = \mathbf{Q}_{D:D+S}, \tag{13}$$

where : here is the index operator.

### 3.5 Loss Functions

We supervise predicted Gaussian means $\mathbf{G}_{:\mu}$ and corresponding class scores $\mathbf{C}$ with Chamfer distance [15] and focal loss [32]

$$\mathcal{L}_{occ} = \text{CD}(\mathbf{G}_{:\mu,0}, \mathbf{P}_g^0) + \sum_{\ell=1}^{L} \text{CD}(\mathbf{G}_{:\mu,\ell}, \mathbf{P}_g^\ell) + \text{FocalLoss}(\mathbf{C}_\ell, \mathbf{C}_g^\ell), \tag{14}$$

where $\text{CD}(\mathbf{G}_0^\mu, \mathbf{P}_g^0)$ encourages initial Gaussian means to capture global pattern of underlying data as pointed out in [53]. For the simplicity of notation, we do not differentiate between static and dynamic Gaussians in Eq. 14.

For box predictions by dynamic Gaussians, $\mathcal{L}_1$ loss is used to supervised box attributes defined in Eq. 5 and focal loss is used to supervise corresponding box class labels

$$\mathcal{L}_{box} = \sum_{\ell=1}^{L} \mathcal{L}_1(\mathbf{G}_\ell^{d,b}, \mathbf{B}_g^\ell) + \text{FocalLoss}(\mathbf{C}_\ell^{d,b}, \mathbf{C}_{g,b}^\ell), \tag{15}$$

where $\mathbb{B}, \mathbf{C}_{g,b}$ denotes ground-truth. Label assignment is done using the Hungarian algorithm [30] during training. Hence, the 3D Loss can be written as

$$\mathcal{L}_{3d} = \mathcal{L}_{occ} + \lambda_{3d}\mathcal{L}_{box}, \tag{16}$$

where $\lambda_{3d}$ is the weighting factor.

For rendered depth and semantic maps from Gaussians at all stages, we supervise depth with $\mathcal{L}_1$ loss and semantics with cross-entropy loss

$$\mathcal{L}_r = \sum_{\ell=1}^{L} \mathcal{L}_1(\hat{D}_\ell, \bar{D}) + \text{CE}(\hat{S}_\ell, \bar{S}), \tag{17}$$

where $\hat{D}, \hat{S}$ are the rendered depth and semantic maps with $\bar{D}, \bar{S}$ being the ground-truth. Here we project LiDAR points with their ground-truth semantic labels that are available from datasets [6, 45] onto each camera view to obtain $\bar{D}, \bar{S}$. Therefore the final loss can be written as

$$\mathcal{L} = \mathcal{L}_{3d} + \lambda\mathcal{L}_r, \tag{18}$$

where $\lambda$ is the weighting factor.

## 4 Experiments

### 4.1 Experiment Setup

**Datasets**: We evaluate our model on the Occ3D benchmark [50] which bootstraps the nuScenes [6] and Waymo-Open [45] dataset.[1] nuScenes consists of 1,000 scenes with a split of $700/150/150$ for training, validation and testing. Occ3D-nuScenes annotates 3D occupancy ground-truth providing 17 semantic classes. The voxel grid range is $[-40\text{m}, -40\text{m}, -1\text{m}, 40\text{m}, 40\text{m}, 5.4\text{m}]$ along the $X, Y$ and $Z$ axis with a grid resolution of $200 \times 200 \times 16$ and voxel size of $0.4\text{m}$. The original image resolution is $900 \times 1600$. Waymo Open [45] has 798 training scenes and 202 validation scenes. Occ3D-Waymo provides 3D semantic occupancy ground-truth of 15 semantic classes with 1 class being *General Object (GO)*. The voxel grid resolution and voxel size is the same as Occ3D-nuScenes. On Waymo, the original image resolution is $1280 \times 1920$ for the front, front-left and front-right cameras. For the side-left and side-right cameras, the original image resolution is $1040 \times 1920$.

**Evaluation Metrics**: We evaluate our model under the mIoU and RayIoU [47] metric:

$$\text{mIoU} = \frac{|P \cap G|}{|P \cup G|}, \quad \text{RayIoU} = \frac{\sum_{r \in \mathcal{R}} |P_r \cap G_r|}{\sum_{r \in \mathcal{R}} |P_r \cup G_r|},$$

where $P, G$ are the set of occupied voxels in prediction and ground-truth respectively. $\mathcal{R}$ is the set of all emulated LiDAR rays, and $P_r, G_r$ are the sets of occupied voxels intersected by ray $r$ in prediction and ground-truth respectively.

**Implementation Details**: We implement our proposed method in PyTorch [42]. Following previous works [35, 53, 4], we use ResNet-50 [20] as image backbone to extract multi-camera image features. On nuScenes, we resize input images to the resolution of $256 \times 704$. On Waymo, all input images are resized and padded to $640 \times 960$. For Ours-tiny, we set number of static Gaussian queries $S = 500$ and number of dynamic Gaussian queries $D = 100$. For Ours-large, we set $S = 4000$ and $D = 800$, respectively. We use $L = 6$ transformer layers to conduct coarse-to-fine prediction. We set $\lambda_{3d} = 0.2$ to balance box loss $\mathcal{L}_{box}$ and occupancy loss $\mathcal{L}_{occ}$. For rendering loss $L_r$, we set $\lambda = 0.05$ for stage $\ell = 1, 6$, and $\lambda = 0.01$ for the rest. We use AdamW [38] as the optimizer with weight decay of $0.01$. We train all our models with an initial learning rate of $2 \times 10^{-4}$ and decays with CosineAnnealing [39] schedule. For experiments on Waymo, we sample 20% of the data matching practices in previous works [53, 50]. Unless otherwise specified, we train all our models with a global batch size of 8 for 100 epochs using NVIDIA A100 GPUs. During inference, we adopt the standard practice and make use of the camera visibility masks provided by the dataset [50] and only evaluate in unoccluded regions. Inference runtime is measured on a single idle A100 GPU with PyTorch `fp32` backend.

---

[1] nuScenes is under a CC BY-NC-SA 4.0 license and Waymo license terms can be found here: `https://waymo.com/open/terms/`.

Table 1: 3D semantic occupancy results on Occ3D-nuScenes validation set [6, 50]. Cons. Veh stands for "Construction Vehicle" and Dri. Sur stands for "Drivable Surface". We note that for fair comparison, both ODG-T and ODG-L here are trained without using future frames. **Bold**/Underline: Best/second best results. *indicates self-supervised methods.

| Method | mIoU | Others | Barrier | Bicycle | Bus | Car | Cons. Veh | Motorcycle | Pedestrian | Traffic Cone | Trailer | Truck | Dri. Sur | Other flat | Sidewalk | Terrain | Manmade | Vegetation | RayIoU | FPS |
|---|---|---|---|---|---|---|---|---|---|---|---|---|---|---|---|---|---|---|---|---|
| RenderOcc [41] | 26.11 | 4.84 | 31.72 | 10.72 | 27.67 | 26.45 | 13.87 | 18.2 | 17.67 | 17.84 | 21.19 | 23.25 | 63.2 | 36.42 | 46.21 | 44.26 | 19.58 | 20.72 | 19.5 | 3.0 |
| GaussRender [10] | 30.38 | 8.87 | 40.98 | 23.25 | 43.76 | 46.37 | 19.49 | 25.2 | **23.96** | 19.08 | 25.56 | 33.65 | 58.37 | 33.28 | 36.41 | 33.21 | 22.76 | 22.19 | 37.5 | - |
| GaussTR* [27] | 12.27 | - | 6.5 | 8.54 | 21.77 | 24.27 | 6.26 | 15.48 | 7.94 | 1.86 | 6.1 | 17.16 | 36.98 | - | 17.21 | 7.16 | 21.18 | 9.99 | - | - |
| SparseOcc (8f) [47] | 30.1 | - | - | - | - | - | - | - | - | - | - | - | - | - | - | - | - | - | 34.0 | 17.3 |
| SparseOcc (16f) [47] | 30.6 | - | - | - | - | - | - | - | - | - | - | - | - | - | - | - | - | - | 35.1 | 12.5 |
| OPUS-T (8f) [53] | 33.2 | 10.72 | 39.82 | 21.27 | 39.76 | 45.25 | 23.41 | 21.80 | 17.81 | 19.26 | 27.48 | 33.20 | 71.61 | 37.12 | 45.13 | 43.59 | 33.80 | 33.18 | 38.4 | **22.4** |
| OPUS-L (8f) [53] | 36.2 | 11.95 | 43.45 | 25.51 | 40.95 | 47.24 | 23.86 | 25.89 | 21.26 | **29.06** | 30.13 | 35.28 | 73.13 | **41.08** | **47.01** | 45.66 | 37.40 | **35.27** | 41.2 | 7.2 |
| GaussianFlowOcc* [4] | 16.02 | - | 7.23 | 9.33 | 17.55 | 17.94 | 4.5 | 9.32 | 8.51 | 10.66 | 2.00 | 11.80 | 63.89 | - | 31.11 | 35.12 | 14.64 | 12.59 | 16.47 | 10.2 |
| ODG-T (8f) | 35.54 | 13.69 | 38.97 | 23.02 | 46.75 | 49.33 | 25.79 | 23.63 | 20.73 | 18.54 | 30.01 | 35.61 | 76.84 | 39.33 | 45.01 | 46.78 | 37.45 | 32.24 | 39.2 | 20.1 |
| ODG-L (8f) | **38.18** | **14.11** | **46.62** | **27.09** | **48.77** | **52.09** | **26.79** | **28.05** | 23.21 | 27.92 | **30.86** | **38.17** | **77.13** | 40.35 | 46.94 | **47.37** | **40.01** | 33.52 | **42.3** | 4.9 |

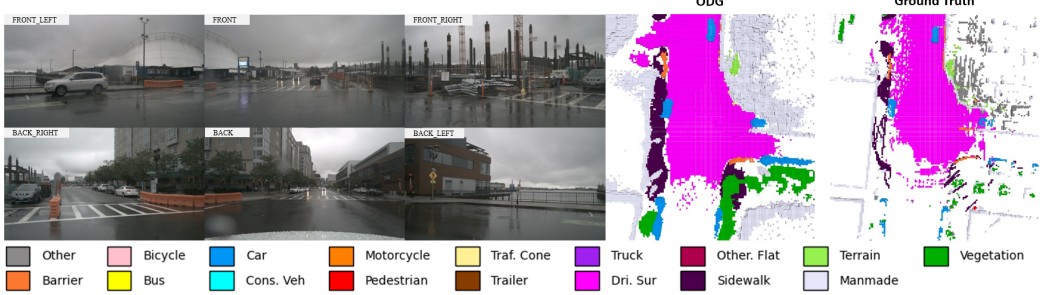

Figure 2: Visualization of ODG prediction on the Occ3D-nuScenes [50, 6] validation set. The ODG can capture all the vehicles on a gloomy rainy day.

## 4.2 Evaluation Results

In this section, we report evaluation results on the Occ3D benchmark [50] and compare with latest state-of-the-art methods.

**nuScenes** Our results on Occ3D-nuScenes is summarized in Tab. 1. One can see that our method achieves new state-of-the-art results in terms of both mIoU and RayIoU, while maintaining competitive inference speed even when compared to latest efficient approaches. Specifically, ODG-T (8f) achieves an mIoU of 35.54 with a RayIoU of 39.2, outperforming OPUS-T (8f) who has an mIoU of 33.2 (-2.34) and a RayIoU of 38.4 (-0.8), while still having an inference runtime of 20.1 FPS. Similarly, ODG-T also easily outperforms both SparseOcc (8f), SparseOcc (16f) and GaussRender with significant margins. Meanwhile, our heavy variant ODG-L sets new best result eventually obtaining an mIoU of 38.18 with a RayIoU of 42.3, surpassing previous best with a definitive margin of +1.98 and +1.1, respectively.

It is worth noting that given our specific design to attend to the dynamic agents in the scene, we show significant improvement when examining the key dynamic object classes. As shown in Tab. 2, for the classes of *Bus, Car, Construction Vehicle (Cons. Veh), Motorcycle*, and *Truck*, ODG-L carries a significant lead of +4.13 for mIoU, once again demonstrating the efficacy of our proposed strategy of handling dynamic agents.

Table 2: Occupancy prediction results over key dynamic object classes on Occ3D-nuScenes [50] validation set. ODG achieves consistent improvement across all dynamic categories. **Bold**/Underline: Best/second best results.

| Method | mIoU | Bus | Car | Cons. Veh | Motorcycle | Truck |
|---|---|---|---|---|---|---|
| GaussRender [10] | 33.69 | 43.76 | 46.37 | 19.49 | 25.2 | 33.65 |
| OPUS-T (8f) [53] | 32.68 | 39.76 | 45.25 | 23.41 | 21.80 | 33.20 |
| OPUS-L (8f) [53] | 34.64 | 40.95 | 47.24 | 23.86 | 25.89 | 35.28 |
| ODG-T (8f) | 36.22 | 46.75 | 49.33 | 25.79 | 23.63 | 35.61 |
| ODG-L (8f) | **38.77** | **48.77** | **52.09** | **26.79** | **28.05** | **38.17** |

Table 3: 3D semantic occupancy results on Occ3D-Waymo validation set [45, 50]. GO stands for "General Object". Traf. Light stands for "Traffic Light" and Cons. Cone stands for "Construction Cone".

| Method | mIoU | GO | Vehicle | Bicyclist | Pedestrian | Sign | Traf. Light | Pole | Cons. Cone | Bicycle | Motorcycle | Building | Vegetation | Tree Trunk | Road | Walkable | RayIoU | FPS |
|---|---|---|---|---|---|---|---|---|---|---|---|---|---|---|---|---|---|---|
| BEVDet [23] | 9.88 | 0.13 | 13.06 | 2.17 | 10.15 | 7.80 | 5.85 | 4.62 | 0.94 | 1.49 | 0.00 | 7.27 | 10.06 | 2.35 | 48.15 | 34.12 | - | - |
| BEVFormer [31] | 16.76 | 3.48 | 17.18 | 13.87 | 5.9 | 13.84 | 2.7 | 9.82 | 12.2 | 13.99 | 0.00 | 13.38 | 11.66 | 6.73 | 74.97 | 51.61 | - | - |
| TPVFormer [25] | 16.76 | 3.89 | 17.86 | 12.03 | 5.67 | 13.64 | 8.49 | 8.90 | 9.95 | 14.79 | 0.32 | 13.82 | 11.44 | 5.8 | 73.3 | 51.49 | - | 4.6 |
| CTF-Occ [50] | 18.73 | 6.26 | 28.09 | 14.66 | 8.22 | 15.44 | 10.53 | 11.78 | 13.62 | 16.45 | 0.65 | 18.63 | 17.3 | 8.29 | 67.99 | 42.98 | - | 2.6 |
| OPUS-L [53] | 19.00 | 4.66 | 27.07 | 19.39 | 6.53 | 18.66 | 6.41 | 11.44 | 10.40 | 12.90 | 0.00 | 18.73 | 18.11 | 7.46 | 72.86 | 50.31 | 24.7 | 8.5 |
| ODG-L | 21.35 | 5.09 | 31.34 | 22.4 | 19.06 | 15.24 | 6.09 | 12.51 | 12.77 | 13.59 | 0.00 | 21.49 | 17.89 | 8.37 | 78.19 | 56.28 | 25.9 | 5.6 |

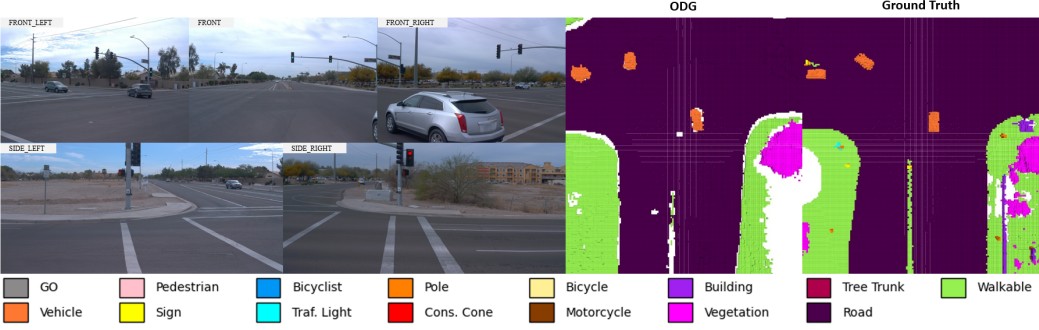

Figure 3: Visualization of ODG prediction on the Occ3D-Waymo [50, 45] validation set.

**Waymo** We further evaluate our ODG on the Occ3D-Waymo dataset and the results are presented in Tab. 3. We note that Occ3D-Waymo is a much less well evaluated occupancy benchmark especially for camera-only methods, given its challenging conditions (*e.g.* almost no view overlap between cameras). We train ODG-L of with 20% of the training data following practice in previous methods [53, 54]. To the best of our knowledge, the most comprehensive source which compiles vision-only methods are from [50, 53], hence we use them for comparison. It is clear from Tab. 3 that our ODG obtains a definitive lead of +2.35 for mIoU and +1.2 for RayIoU when compared to OPUS. Upon further examining Tab. 3 one can see that ODG establishes far surpass the second-best under class *Vehicle(+3.25), Bicyclist(+3.01)* and *Pedestrians(+8.91)*, which are all critical traffic agents, once again proving the superiority of our modeling of dynamic objects.

### 4.3 Visualization

We showcase visualizations of predicted occupancy of ODG on nuScenes and Waymo in Fig. 2 and Fig. 3. In Fig. 2, under gloomy rainy weather where the ground-truth is sparse especially in object-centric categories (*e.g.* cars in this case), ODG effectively outputs dense predictions capturing all the cars in the scene. In Fig. 3, ODG also captures all the traffic agents. We attribute this attractive behavior to our dynamic Gaussian queries dedicated to modeling the dynamic scene agents.

### 4.4 Ablation Studies

In this section, we conduct multiple ablation studies to analyze the effects of various components in our proposed ODG. For all our ablation studies, we adopt ODG-T and train on the Occ3D-nuScenes for 24 epochs.

**Temporal Alignment** As pointed in Sec. 3.3, in order to sample features correctly from the history frames, it is critical to move the Gaussians according to their respective motions before projecting them onto image feature plane. Hence, here we study the effects of performing different type of motion compensation. As shown in Tab. 5b, if we merely correct ego-motion (camera-motion) for the dynamic Gaussians, both mIoU and RayIoU suffers a noticeable drop of -0.71 and -0.5 respectively. This demonstrates it is essential to compensate object motion for dynamic agents.

**Query Interaction** Given we have dynamic and static Gaussians modeling the scene, it is important to make them aware of each other. We analyze the effect of different attention mechanisms in Tab. 5a. In our experiments, we first tried performing cross attention with dynamic query features serving as queries, and static query features as keys and values, which gave us a modest improvement. A more straight-forward way is to concatenate the dynamic and static query features together, and perform self attention on top of concatenated features. This lead to further better results as shown in Tab. 5a.

Table 4: Impact of different components inside ODG on model performance.

| Motion compensation | Query attention | Rendering Sup | mIoU | RayIoU$_{1m}$ | RayIoU$_{2m}$ | RayIoU$_{4m}$ |
|:---:|:---:|:---:|:---:|:---:|:---:|:---:|
| | | | 30.80 | 27.9 | 36.6 | 42.0 |
| ✓ | | | 31.78 | 28.7 | 37.3 | 42.6 |
| ✓ | ✓ | | 32.13 | 29.3 | 37.8 | 43.1 |
| ✓ | ✓ | ✓ | 32.82 | 31.1 | 40.5 | 43.8 |

Table 5: Ablation studies on components related to dynamic Gaussian queries.

(a) Effects of Query Attention.

| Query Attention | mIoU | RayIoU |
|:---:|:---:|:---:|
| Cross Attn | 31.95 | 36.3 |
| Self Concat Attn | 32.13 | 36.7 |

(b) Effects of Motion Compensation.

| Ego Comp. | Dyn. Comp | mIoU | RayIoU |
|:---:|:---:|:---:|:---:|
| ✓ | | 31.17 | 35.7 |
| ✓ | ✓ | 31.78 | 36.2 |

We posit that running self attention on all features in an exhaustive manner makes all queries become aware of each other therefore facilitating more information flow, leading to improved results.

**Rendering Supervision** To reduce spatial artifacts through projective constraints, we leverage 3D Gaussian Splatting and render depth and semantic maps to each camera view. As shown in Tab. 4, this resulted in a noticeable improvement both in mIoU and RayIoU, with mIoU+0.69 and RayIoU+0.70. By injecting the auxiliary supervision signals from LiDAR points projected onto each camera view (keyframes only, constrained by sensor calibration), it effectively enforces geometric and semantic consistency especially during the early stages of rendering, which helps the model learn more effectively for the regions that the 3D occupancy ground-truth are ambiguous or noisy. It is worth noting that during inference, we turn off rendering hence incurring no extra computation cost, reaping the benefit with zero increase in inference latency.

We summarize the effect of the different components in our proposed method in Tab. 4. It is evident that by progressively enabling different modules in ODG, the model performs increasingly well, validating the soundness of the designs that we incorporated into our system.

## 5 Conclusions and Discussions

In this paper, we present ODG, a novel approach to 3D occupancy prediction based on dual Gaussian queries. ODG defines two distinctive sets of queries consisting of dynamic and static Gaussian queries, aiming to better model dynamic scene agents. To make ODG attend to moving objects, we expand the standard 3D Gaussian properties of dynamic queries with 3D bounding box attributes, which effectively guides queries to dedicate resource to capture complex scene dynamics. Self attention is utilized to establish connection between dynamic and static queries. To enable sufficient model capacity, ODG adopts a coarse-to-fine prediction paradigm when it comes to predicting the Gaussian parameters, which allows a large number of Gaussians to be utilized. To reduce spatial artifacts in 3D occupancy, we enforce projective constraints through rendering depth and semantic maps to each camera view leveraging 3D Gaussian Splatting, supervised by projected LiDAR points. Our extensive experiments on the Occ3D-nuScenes and Occ3D-Waymo benchmark demonstrates ODG sets new state-of-the-art results while maintaining highly competitive efficiency.

**Limitations.** However, as promising as ODG is, it does not come without limitations. Readers might have observed that currently the Gaussian parameters other than mean (namely scale, rotation and opacity) is only optimized through rendering loss, rather than for instance, aggregating nearby Gaussians to get occupancy and learn from occupancy ground-truth as well, albeit such aggregation incurs significant cost. Therefore exploring ways to improve optimization of Gaussians would be an interesting research direction.

**Broader Impacts.** Our proposed ODG provides more accurate 3D occupancy prediction while maintaining inference efficiency, which is beneficial to safe, energy-efficient autonomous driving. Furthermore, given the versatility and efficiency of our ODG, we think it is a promising venue towards building a unified perception and prediction module for autonomous driving, which will improve the homogeneity of the entire stack that will in turn lower overall operating cost, yielding economic benefit. On the other hand, during the development of this work, several experiments did not make it into the paper (*e.g.* due to lack of time), which incurred extra energy consumption.

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

# Appendix

We provide more experiment analysis regarding various aspects of ODG.

## A. Coarse-to-fine Prediction of Gaussian Parameters

We present details of the coarse-to-fine refinement used in ODG. For notation simplicity, we denote motion compensation, feature sampling, cross and self attention, along with dual query-attention presented in Figure 1 as a single layer $\hat{\mathcal{T}}_l$, rest of the notations are the same as defined in Section 3.2. To further enhance clarity, since coarse-to-fine refinement doesn't apply to bounding box attributes (i.e. $K_0 = 1$ throughout all the layers $\hat{\mathcal{T}}_l$), we have omitted them in Algorithm 1 below.

**Input:** images features $\mathbf{F}$; static Gaussian queries $\{\mathbf{G}_0^s, \mathbf{Q}_0^s\}$; dynamic Gaussian queries
$\{\mathbf{G}_0^d, \mathbf{Q}_0^d\}$;
**Output:** refined Gaussian queries $\{\mathbf{G}_L^s, \mathbf{Q}_L^s\}$, class predictions $\mathbf{C}_L^s$; refined dynamic Gaussian
queries $\{\mathbf{G}_L^d, \mathbf{Q}_L^d\}$, class predictions $\mathbf{C}_L^d$.
**Function** *ODGRefine(*$\mathbf{F}, \mathbf{G}_0^s, \mathbf{Q}_0^s, \mathbf{G}_0^d, \mathbf{Q}_0^d$*)*
    $\mathbf{G}_{:\mu,0}^s \in \mathbb{R}^{S \times K_0 \times 3} \leftarrow \mathcal{U}(0,1), K_0 = 1$
    $\mathbf{G}_{:\mu,0}^d \in \mathbb{R}^{D \times K_0 \times 3} \leftarrow \mathcal{U}(0,1), K_0 = 1$
    // initialize Gaussian means with uniform distribution; rest of the Gaussian parameters left uninitialized
    **for** $l \leftarrow 1$ **to** $L$ **do**
        # static
        $\mathbf{G}_{:\mu,l}^s, \mathbf{Q}_l^s, \mathbf{C}_l^s = \hat{\mathcal{T}}_l(\mathbf{G}_{:\mu,l-1}^s, \mathbf{Q}_{l-1}^s, \mathbf{C}_{l-1}^s)$
        $\mathbf{G}_{\mu:,l}^s = \Phi(\mathbf{G}_{:\mu,l}^s, \mathbf{Q}_l^s)$ // rest of Gaussian params predicted by MLP
        // $\mathbf{G}_{:\mu,l}^s \in \mathbb{R}^{S \times K_l \times 3}, \mathbf{G}_{\mu:,l}^s \in \mathbb{R}^{S \times K_l \times 10}, \mathbf{C}_l^s \in \mathbb{R}^{S \times K_l \times C}, K_{l-1} < K_l$ # coarse-to-fine

        # dynamic
        $\mathbf{G}_{:\mu,l}^d, \mathbf{Q}_l^d, \mathbf{C}_l^d = \hat{\mathcal{T}}_l(\mathbf{G}_{:\mu,l-1}^d, \mathbf{Q}_{l-1}^d, \mathbf{C}_{l-1}^d)$
        $\mathbf{G}_{\mu:,l}^d = \Phi(\mathbf{G}_{:\mu,l}^d, \mathbf{Q}_l^d)$ // rest of Gaussian params predicted by MLP
        // $\mathbf{G}_{:\mu,l}^d \in \mathbb{R}^{D \times K_l \times 3}, \mathbf{G}_{\mu:,l}^d \in \mathbb{R}^{D \times K_l \times 10}, \mathbf{C}_l^d \in \mathbb{R}^{D \times K_l \times C}, K_{l-1} < K_l$
        # coarse-to-fine
    **end**
    **return** $\{\mathbf{G}_L^s, \mathbf{Q}_L^s, \mathbf{C}_L^s\}, \{\mathbf{G}_L^d, \mathbf{Q}_L^d, \mathbf{C}_L^d\}$

**Algorithm 1:** Coarse-to-fine refinement in ODG.

## B. Runtime Efficiency

We profiled ODG-L at inference time with DeepSpeed [43]. Results are summarized in Table 6a below.

Table 6: Runtime analysis of ODG-L (FP32).

(a) Runtime of different components in ODG-L

| Component | Runtime (ms) | Percentage |
|---|---|---|
| img_backbone (ReNet50) | 33.82 | 16.58% |
| img_neck (FPN) | 9.83 | 4.82% |
| ODG-L transformer | 160.32 | 78.59% |

(b) Runtime of different components in ODG-L transformer.

| Component | Runtime (ms) | Percentage |
|---|---|---|
| Self-attention | 70.77 | 44.14% |
| Point sampling | 25.88 | 16.14% |
| Cross-attention | 24.06 | 15.01% |
| FFN | 8.55 | 5.33% |
| . . . | . . . | . . . |

We can see that the transformer part takes up most of the inference cost in ODG-L. We provide further profiling results on ODG-L transformer in Table 6b above. Evidently query self-attention takes up almost half the runtime. One straight-away optimization can be replacing self-attention with efficient attention schemes such as linear attention [55] or state space models (SSMs) [18]. We plan to look into this as part of future work.

## C. Evaluation of Prediction from Each Layer

To provide further insight of the coarse-to-fine scheme in ODG, we evaluate predictions of all 6 layers of ODG-T against ground-truth occupancy. The results are summarized in Table 7 below. In the beginning since the prediction is too coarse, the model behaves poorly in terms of mIoU and RayIoU. Then with our coarse-to-fine refinement, ODG starts gradually learning the scene and gives progressively better results, which validates the effectiveness of our coarse-to-fine refinement design.

Table 7: Evaluation of each layer's prediction in ODG-T.

| ODG-T | # Predicted Gaussians | mIoU | $RayIoU_{1m}$ | $RayIoU_{2m}$ | $RayIoU_{4m}$ | RayIoU |
|---|---|---|---|---|---|---|
| layer 1 | 600 | 0.35 | 1.9 | 2.8 | 3.8 | 2.8 |
| layer 2 | 2400 | 3.17 | 9.5 | 13.5 | 16.3 | 13.0 |
| layer 3 | 9600 | 18.68 | 25.9 | 32.6 | 36.6 | 31.7 |
| layer 4 | 12800 | 26.48 | 29.9 | 37.9 | 42.2 | 36.6 |
| layer 5 | 38400 | 31.02 | 30.5 | 39.9 | 43.3 | 37.9 |
| layer 6 | 76800 | 32.82 | 31.1 | 40.5 | 43.8 | 38.5 |

## D. Ablation on Supervision Signals

In Table 8, we show how ODG performs when there is only occupancy labels available. Compared to full supervision with bbox and rendering labels, ODG suffers a slight drop in terms of mIoU and rayIoU, but still delivers strong performance, which validates our design.

Table 8: Ablation on supervision signal.

| Supervision | mIoU | rayIoU |
|---|---|---|
| Occupancy only | 31.46 | 36.1 |
| Occupancy & bbox & rendering supervision | 32.82 | 38.5 |

## E. Effect of Query Composition

We study the effect of utilizing different types of queries. The results are summarized in Table 9.

Table 9: Effect of query composition on model performance.

| Method | # Queris | mIoU | RayIoU | FPS |
|---|---|---|---|---|
| OPUS-T [53] | 600 (static) | 33.2 | 38.4 | 22.4 |
| OPUS-L [53] | 4800 (static) | 36.2 | 41.2 | 7.2 |
| ODG-T | 500 (static)+100 (dynamic) | 35.5 | 39.2 | 20.1 |
| ODG-L | 4000 (static)+800 (dynamic) | 38.2 | 42.3 | 4.9 |
| ODG-T* | 600 (dynamic) | 34.8 | 38.9 | 17.6 |
| ODG-L* | 4800 (dynamic) | 37.6 | 42.1 | 3.7 |

One can see that by taking into account object motion, ODG definitely outperforms baseline OPUS [53] which only considers ego motion. Interestingly, when we treat all our queries as dynamic (denoted as ODG-*), it still performs better than OPUS. We attribute this gain to the object detection task present in our system which effectively improves dynamic object predictions, and through rendering constraint we also learn better 3D geometry.

