# OpenReview forum: "ODG: Occupancy Prediction Using Dual Gaussians"
_NeurIPS.cc/2025/Conference — NeurIPS 2025 poster_

### Official Review · Reviewer_XGZs · 2025-06-22

**Clarity:** 2
**Significance:** 3
**Originality:** 3
**Rating:** 4
**Confidence:** 4

**Summary:**

This paper tackles 3D occupancy prediction through a detr-like query-based approach. The paper proposes using two sets of queries: one set handling the static environment, and another set handling the dynamic objects. These two types of queries are handled differently, where the static environment is ego-motion compensated over time, and the dynamics predict their own speed to also compensate for object movement.
They further utilise a hierarchical approach where the number of Gaussians increases progressively through the transformer-based neural network. They further leverage Gaussian splatting techniques to use rendering as an auxiliary task.
The paper shows strong performance on both nuScenes and Waymo datasets.

**Questions:**

- Is it possible to show that the network learns how to disentangle the scene?
- Can the velocity estimations be evaluated and visualised?
- Can the effects of the paper's design choices be further validated?

**Ethical Concerns:**

["NO or VERY MINOR ethics concerns only"]

**Final Justification:**

Adding the evaluation on intermediate layers as the resolution increases highlights the models capability to incrementally refine the occupancy. I also appreciate the experiment on understanding the velocity of moving objects and robustness to stochasticity in the training process and think these improves the paper substantially.

While not ground-breaking, this paper makes a relevant contribution to the field.

**Limitations:**

Yes. Could probably add some visualizations of failure cases to sup.mat.

**Quality:**

2

**Strengths And Weaknesses:**

_Stengths_
- Decomposition via Query Types: The paper uses a disentanglement of the scene into static and dynamic components via separate query sets. It is intuitive and aligns well with the physical structure of driving environments.
- Hierarchical Occupancy: The use of progressively increasing Gaussians through the transformer layers is an interesting design choice that (I suppose) incrementally improves the resolution further down in the network. Seems intuitive that this builds a bias in the network to focus on the overall structure first and then focus on the fine-grained details.
- Auxiliary Rendering Task: Incorporating Gaussian rendering as an auxiliary supervision signal adds an extra layer of regularization and interpretability, further improving the model’s structure.
- Strong Empirical Evaluations: The method achieves competitive performance on both nuScenes and Waymo datasets, suggesting good generalizability across benchmarks. Also show performance on both IoU and RayIoU.


_Weaknesses_
- Lack of visualisations for Decomposition: Given that the static/dynamic decomposition is a central contribution, it would be valuable to include qualitative visualisations demonstrating how well the model separates and models each component. Is it possible to visualize the scene given the static queries alongside the dynamic queries?
- Lack of evaluation of hierarchical occupancy: Nothing verifies that the network actually starts with focusing on high-level occupancy and then refines it through the transformer layers. Showing metrics or visualisations from different layers would improve the understanding of the behaviour of the model.
- Query Efficiency Comparison Missing: It would strengthen the paper to compare the number of queries used with other query-based occupancy prediction methods. This would contextualise the efficiency and scalability of the proposed approach. Furthermore, would it benefit the paper to show that it’s the separation between static and dynamic that make the method work well. For instance, what happens if all queries (the static ones too) are treated like the dynamic queries?
- Velocity estimation evaluation: Since learning how to compensate for the moving objects it would be very interesting to evaluate (both quantitatively and qualitatively) how well these match with the ground truth. Understanding how well the model learns velocity/flow would significantly increase my takeaways from the paper.
- Inconsistent Metric Reporting: For instance, Table 4 lacks mean RayIoU over thresholds, which is used elsewhere. Consistent metric reporting across all tables would make interpretation clearer and comparison more fair.
- Ablation Limitations: The ablation studies do not fully disentangle the contributions of rendering supervision, motion compensation, and attention mechanisms. For instance, an ablation with only rendering supervision (without motion compensation or query attention) would help isolate its specific contribution.
It’s unclear how stable the dynamic motion compensation is (the improvement is rather close to variations due to training noise in Table 5). Does the learned behaviour vary significantly across training runs? Reporting results across multiple seeds would clarify this.
An analysis of how the number of temporal input frames affects performance would clarify the model’s temporal robustness.
- Writing and notation polishing: The writing feels a bit rushed and could need some extra polishing. The equation and notations could also be improved. For instance, are (12)-(13) really necessary? Some definitions missing: P is not defined in eq (16)....

---

> ### Author Rebuttal · Authors · 2025-07-30
>
> We thank reviewer XGZs for the constructive feedbacks. We address the reviewer's questions and comments below.
>
> $\textbf{Visualize the scene with static and dynamic queries prediction alongside each other}$:
>
> We have prepared such visualizations and will include them in the final version to further demonstrate the superiority of our proposed ODG. Unfortunately, this year's NeurIPS rebuttal format does not support uploading a PDF for the rebuttal and as such, we are not able to share visualization results during the rebuttal stage.
>
> $\textbf{Metrics of predictions from each layer}$:
>
> We evaluated predictions of all 6 layers of ODG-T against ground-truth occupancy and computed metrics. The results are summarized in Table 1 below. We can clearly see that in the beginning since the prediction is too coarse, the model behaves poorly in terms of mIoU and RayIoU. Then with our coarse-to-fine refinement, ODG starts gradually learning the scene and gives progressively better results, which validates the efficacy of our coarse-to-fine refinement design.
>
> **Table 1. Evaluation of predictions of all 6 layers of ODG-T on nuScenes.**
> | ODG-T    | #Predicted Gaussians | mIoU   | $\text{RayIoU}_{1m}$ |  $\text{RayIoU}_{2m}$  | $\text{RayIoU}_{4m}$ | $\text{RayIoU}$ |
> | -------- | ------- | -------- | ------- | -------- | ------- | ------- |
> | layer 1 | 600    | 0.35  | 1.9    | 2.8    | 3.8    | 2.8    |
> | layer 2 | 2400     | 3.17 |   9.5  | 13.5  | 16.3  | 13.0    |
> | layer 3   | 9600    | 18.68    | 25.9 | 32.6   | 36.6  | 31.7  |
> | layer 4   | 12800    |  26.48   | 29.9    | 37.9  | 42.2 | 36.6   |
> | layer 5   | 38400  | 31.02  | 30.5  | 39.9  | 43.3   | 37.9  |
> | layer 6   | 76800  | 32.82    | 31.1    | 40.5  | 43.8  | 38.5  |
>
> $\textbf{Query efficiency comparison}$:
>
> We compare with OPUS [1] which is a SotA query-based method for occupancy. The results are summarized in Table 2 below. We note that to ensure a fair comparison, the results of OPUS are directly obtained from their paper. It is clear that with the same number of total queries, ODG outperforms OPUS with significant margin, while only incurring a minimal overhead. For treating all queries as dynamic (denoted as ODG* in Table 2), we can see that even though ODG experienced a minor drop in terms of mIoU and RayIoU, it still definitively outperformed baseline OPUS. We attribute this gain to the object detection task present in our system which effectively improves dynamic object predictions, and through rendering constraint we also learn better 3D geometry.
>
> **Table 2. Query efficiency comparison.**
> | method    | #queries | mIoU    | RayIoU | FPS
> | -------- | ------- | -------- | ------- | ------- |
> | OPUS-T  | 600 (static)  | 33.2 | 38.4  | 22.4
> | OPUS-L | 4800 (static)   |  36.2 | 41.2   | 7.2
> | ODG-T | 500 (static) + 100 (dynamic)  |  35.5 | 39.2  | 20.1
> | ODG-L | 4000 (static) + 800 (dynamic) |  38.2  | 42.3  | 4.9
> | ODG-T* | 600 (dynamic)  |  34.8 | 38.9  | 17.6
> | ODG-L* | 4800 (dynamic) |  37.6  | 42.1  | 3.7
>
> [1] Wang, Jiabao, et al. "Opus: occupancy prediction using a sparse set." Advances in Neural Information Processing Systems 37 (2024): 119861-119885.
>
> $\textbf{Velocity estimation evaluation}$:
>
> Since our velocity is approximated by ego vehicle at different image timestamps and there is no velocity ground-truth, hence direct quantitative evaluation is not feasible in this case. However, we validate the effectiveness of the dynamic queries predictions through the following surrogate experiment on nuScenes: given current timestamp $t$ (keyframe), we denote occupancy ground-truth $O_t$ and predictions $O_t^p$. We denote the immediate previous timestamp $\hat{t}$ (also keyframe, where occupancy GT is available) and its occupancy GT $O_{\hat{t}}$ and predictions $O_{\hat{t}}^p$. Then we warp prediction at current time $O_t^p$ to previous time using ego-motion for static queries and velocity for dynamic queries and denote as $O_{t\rightarrow\hat{t}}^p$. We evaluation both $O_{\hat{t}}^p$ and $O_{t\rightarrow\hat{t}}^p$ against ground-truth $O_{\hat{t}}$ at time $\hat{t}$. The results are summarized in Table 3 below:
>
> **Table 3. Surrogate experiment of evaluating velocity estimation.**
> | Pred    | GT | mIoU    | RayIoU |
> | -------- | ------- | -------- | ------- |
> | $O_t^p$  | $O_t$  |  32.82 | 38.5  |
> | $O_t^p$|  $O_{\hat{t}}$  | 26.77 |  36.1  |
> | $O_{t\rightarrow\hat{t}}^p$  | $O_{\hat{t}}$  |  30.74 |  37.4  |
>
> We can clearly see that due to motion misalignment if we directly evaluate prediction $O_{t}^p$ against GT $O_{\hat{t}}$ at previous time, there is a significant degradation in terms of mIoU and RayIoU. But if we warp current prediction $O_t^p$ to previous time $O_{t\rightarrow\hat{t}}^p$ using estimated velocities of the queries, such misalignment is corrected and we obtain significantly improved results. This demonstrates the usefulness of our estimated velocity.
>
> We note that we will add velocity flow visualization in the final version, given during the rebuttal stage it is not feasible for us to share any type of rich media such as images and videos.
>
> $\textbf{Ablation on only using rendering supervision}$:
>
> We conducted such ablation experiment with ODG-T (trained for 24 epochs on nuScenes, consistent with Table 4 in main paper) and the results are summarized in Table 4 below:
>
> **Table 4. More ablation studies of ODG-T including only using rendering supervision. Trained for 24 epochs on nuScenes.**
> | Motion compensation    | Query Attention | Rendering Sup | $\text{mIoU}$ | $\text{RayIoU}_{1m}$ | $\text{RayIoU}_{2m}$ | $\text{RayIoU}_{4m}$ | $\text{RayIoU}$
> | -------- | ------- | -------- | -------- | -------- | -------- | -------- | -------- |
> |   |     |  | 30.80 | 27.9 | 36.6 | 42.0 | 35.5 |
> |  |     | yes | 31.29 | 28.3 | 36.9 | 42.2 | 35.8|
> | yes |     |  | 31.78 | 28.7 | 37.3 | 42.6 | 36.2 |
> | yes  | yes    |  | 32.13 | 29.3 | 37.8 | 43.1 | 36.7
> | yes  | yes    | yes | 32.82 | 31.8 | 41.2 | 44.5 | 38.5 |
>
> we can see that even with only rendering supervision, our method is able to achieve a modest improvement, validating the effectiveness of our design.
>
> $\textbf{Inconsistent metric reporting on RayIoU}$:
>
> We have corrected such inconsistency by adding $\text{RayIoU}$ as last column in Table 4 in the main paper. The changes will be reflected in the final version.
>
> $\textbf{Stableness of improvement coming from dynamic motion compensation}$:
>
> We launched another three separate experiments with distinctively different seeds to validate the stability of improvement. The results are summarized in Table 5 below. We can see that the metrics yielded across runs with different seeds are very stable, demonstrating the validity of the improvement from dynamic motion compensation. To facilitate reproducing, experiments were conducted using PyTorch 2.1.2, NVIDIA Driver 535.161.08 with CUDA 12.1 on A100 GPUs.
>
> **Table 5. Multiple experiments of ODG-T with different seeds to validate stableness of improvement coming from dynamic motion compensation. Trained for 24 epochs on nuScenes.**
> | Seed    | Ego Comp | Dyn. Comp  | mIoU | RayIoU |
> | -------- | ------- | ------- | ------- | ------- |
> | 1024 (paper)  |  yes |  |  31.17   |  35.7   |
> | 1024 (paper) | yes    | yes  | 31.78    | 36.2
> | 42   |  yes |   | 31.17   |  35.7   |
> | 42 |  yes  | yes | 31.78    | 36.2
> | 3407 |  yes  | | 31.18  | 35.7     |
> |  3407  | yes  | yes   | 31.80   | 36.2
> | 2025  | yes   |  | 31.17   |  35.7   |
> | 2025  | yes  | yes | 31.78    | 36.2
>
> $\textbf{Writing and notation polishing}$:
>
> We have condensed Eq.12-13 into a single-line text description and added the definition of $P$ in Eq. 16. The changes will be reflected in the final version.

---

### Official Review · Reviewer_M77i · 2025-06-22

**Clarity:** 3
**Significance:** 2
**Originality:** 2
**Rating:** 4
**Confidence:** 4

**Summary:**

The paper proposes ODG, a method that employs a hierarchical dual sparse Gaussian representation for 3D occupancy prediction. It explicitly separates dynamic and static queries, and refines Gaussian parameters in a coarse-to-fine manner. Notably, the approach leverages supervision from 3D bounding boxes through a box encoder, as well as 2D labels via a rendering process. These innovations lead to strong performance on the Occ3D benchmark.

**Questions:**

- Does each dynamic query correspond to a single semantic class? If not, applying motion compensation may be questionable. If so, how are queries handled when they fail to match any 3D bounding box during the Hungarian matching process?
- How would the model perform if only occupancy labels were used for supervision?
- [minor] In Table 1, the best and second-best results are not correctly highlighted.

**Ethical Concerns:**

["NO or VERY MINOR ethics concerns only"]

**Final Justification:**

Although some high-level concepts are similarly presented in related works, the paper introduces its own novel designs. The experiments included in the rebuttal demonstrate that ODG can perform well without utilizing extra labels. Therefore, I raise the rating into borderline accept.

**Limitations:**

Yes

**Quality:**

2

**Strengths And Weaknesses:**

Strengths:
- The writing and figures in paper are clear.
- The paper presents extensive experimental results, demonstrating their high accuracy and fast inference speed.

Weaknesses:
- The novelty of the proposed model is limited. The separation of dynamic and static modeling is straightforward, and the coarse-to-fine refinement mechanism has already been explored in prior works [3, 4]. Additionally, the use of supervision via rendering shares conceptual similarities with approaches such as RenderOcc [1].
- The method leverages extra supervision signals, including 3D bounding boxes and 2D image labels. As a result, the comparison with prior methods that do not utilize such annotations may be unfair.

[1] RenderOcc: Vision-Centric 3D Occupancy Prediction with 2D Rendering Supervision
[2] Fully Sparse 3D Panoptic Occupancy Prediction
[3] Opus: occupancy prediction using a sparse set

---

> ### Author Rebuttal · Authors · 2025-07-30
>
> We thank reviewer M77i for the constructive feedbacks. We address the reviewer's questions and comments below.
>
> $\textbf{Contributions on dynamic and static modeling:}$
>
> Designing an occupancy prediction network that properly considers both dynamic and static objects is not trivial and requires several careful designs. First, it is critical to create a proper learning scheme for dynamic queries, as velocity ground truth is not available. Without proper motion modeling and learning, one cannot correctly aggregate information across time steps. In addition, it is also important to introduce some form of feature interaction between dynamic and static queries in an efficient way, without which the occupancy prediction performance would not be good and/or significant additional computation cost would be incurred. Moreover, our explicit 3D representation using 3D Gaussians also makes it more efficient and feasible to incorporate motion of dynamic objects, as compared to commonly used dense representations like birds-eye-view or 3D voxels which would require expensive 2D/3D warping.
>
> $\textbf{Comparing to OPUS and SparseOcc}$:
>
> While OPUS [1] and SparseOcc [2] leverage some forms of coarse-to-fine design, their models miss certain important factors and/or require some cumbersome designs. For instance, OPUS is a query-based method using points as scene representation, which cannot directly be used for rendering. In addition, OPUS does not consider the motion of dynamic objects at all, which makes the modeling incomplete as there are almost always dynamic objects in a driving scene. SparseOcc employs a series of sparse voxel decoders to filter out empty grids and predict occupancy state of retained voxels, which is inefficient. In contrast, we predict 3D Gaussians in a streamlined fashion without any pre-defined locations and takes dynamic objects into account. Our proposed ODG introduces a new design offering more complete modeling, higher efficiency, as well as the possibility of leveraging additional supervision, which goes beyond incremental improvements.
>
> [1] Wang, Jiabao, et al. "Opus: occupancy prediction using a sparse set." Advances in Neural Information Processing Systems. 2024
>
> [2] Liu, Haisong, et al. "Fully sparse 3d occupancy prediction." European Conference on Computer Vision. 2024.
>
> $\textbf{Difference from RenderOcc}$:
>
> While RenderOcc also leverages neural rendering to improve occupancy prediction, it is NeRF-based which carries a significantly higher cost in terms of training time and memory required. In addition, RenderOcc adopts BEVStereo4D [1] as feature representation, which incurs a much higher inference runtime as shown in Table 1 in our paper. In contrast, in ODG our query-based representation enables a much more efficient scene representation, while also leveraging Gaussian splatting resulting in much faster rendering and significantly reduced training and inference cost.
>
> [1] Li, Yinhao, et al. "Bevstereo: Enhancing depth estimation in multi-view 3d object detection with temporal stereo." Proceedings of the AAAI Conference on Artificial Intelligence. Vol. 37. No. 2. 2023.
>
> $\textbf{Leveraging additional supervision signals}$:
>
> We would like to point out that being able to efficiently leverage additional supervision signals (when available) is part of our contributions. Previous works either cannot incorporate these supervisions, e.g., OPUS [1] does not support rendering, or does it in a very expensive way, e.g., RenderOcc uses volume rendering which is both computationally and memory expensive. Moreover, many of these existing works do not model dynamic objects, making it not possible to properly apply the extra supervisions like rendering loss.
>
> Also, we note that we do not use more ground-truth information as compared to other recent occupancy prediction methods. In order to create the occupancy labels, 3D object segmentation ground truths are used (see Occ3D [3]) which is actually more expressive than 3D bounding boxes. We obtain 2D sparse depth and semantic maps by projecting from LiDAR and the ground-truth 3D segmentation, which have also been used to create occupancy labels in existing Occ3D benchmark.
>
> [1] Wang, Jiabao, et al. "Opus: occupancy prediction using a sparse set." Advances in Neural Information Processing Systems. 2024
>
> [2] Liu, Haisong, et al. "Fully sparse 3d occupancy prediction." European Conference on Computer Vision. 2024.
>
> [3] Tian, Xiaoyu, et al. "Occ3D: A Large-Scale 3D Occupancy Prediction Benchmark for Autonomous Driving." Advances in Neural Information Processing Systems. 2023
>
> $\textbf{Whether each dynamic query corresponds to a single semantic class}$:
>
> We do not impose such a constraint explicitly in the model. However, each dynamic query is implicitly encouraged to capture one object in our learning framework. This is because when motion compensation is applied to a dynamic query, the corresponding set of points move the same way, and we perform motion compensation for multiple temporal steps. In this way, the set of points corresponding to the same query learn to match with the same dynamic object in the scenes, as they always move together. We can provide a visualization of this in the revised paper (unfortunately this year's NeurIPS does not support uploading a PDF for rebuttal).
>
> $\textbf{If only occupancy labels are used for supervision}$:
>
> We conducted experiment with ODG-T and trained for 24 epochs on nuScenes where only occupancy labels are used in training. The results are listed in Table 1 below. We can see when only using occupancy labels as supervision, ODG-T experienced a minor drop in performance but is still performing respectably well, validating the robustness of our designs.
>
> **Table 1. Ablation study on different supervision signals with ODG-T on nuScenes**
> | Supervision    | mIoU | RayIou |
> | -------- | ------- | ------- |
> | occupancy only  |  31.46   | 36.1
> | occupancy \& bbox \& rendering supervision |  32.82  | 38.5
>
> $\textbf{Highlighting error in Table 1}$:
>
> We have corrected the errors and the changes will be reflected in the final version.

---

> > ### Comment · Reviewer_M77i · 2025-08-07
> >
> > Sorry for the late response. Most of my concerns have been addressed. I would like to raise my rating.
> >
> > I recommend that the authors include visualizations of the dynamic query in their revised draft. Furthermore, it would enhance comparison with previous methods if the authors could present ODG with occupancy and rendering supervision (rendering supervision is suitable as it operates similarly to self-supervision) in Table 1. In my opinion, this approach would create a more compelling narrative: ODG achieves state-of-the-art results without extra labels and retains the potential to use 2D labels for further enhancement.

---

> > > ### Author Response · Authors · 2025-08-08
> > >
> > > Dear Reviewer M77i,
> > >
> > > Thank you for recognizing our efforts in addressing your concerns! We truly appreciate your constructive feedback. Your insights were invaluable, not only for improving the manuscript but also for giving us the opportunity to reflect more deeply on our research.
> > >
> > > Best,
> > >
> > > Authors

---

### Official Review · Reviewer_JcRJ · 2025-07-02

**Clarity:** 3
**Significance:** 3
**Originality:** 2
**Rating:** 4
**Confidence:** 3

**Summary:**

The article introduces ODG (Occupancy Prediction Using Dual Gaussians), a novel method for occupancy prediction in autonomous driving. This approach integrates 3DGS and models the scene by dividing it into two categories—static and dynamic—in a hierarchical manner. It employs a gaussian transformer to predict gaussian parameters and leverages the rendering capability of 3DGS to obtain supervisory signals. The method achieves SOTA performance across multiple benchmarks.

**Questions:**

1. ODG-L achieves better accuracy but at the cost of significantly reduced inference speed (4.9 FPS vs. 20.1 FPS). Can the authors provide more insight into what components contribute most to this slowdown? Are there practical ways to improve the efficiency of ODG-L without sacrificing too much performance?
2. The ablation studies (Sec. 4.4) are conducted after training for only 24 epochs, while the final model is trained for 100 epochs. Could the conclusions drawn from these ablations change when evaluated over longer training periods? For example, do certain components become more or less important with extended training?

**Ethical Concerns:**

["NO or VERY MINOR ethics concerns only"]

**Final Justification:**

The authors have fully addressed my concerns through a clear algorithmic description of the coarse-to-fine mechanism, detailed efficiency profiling, and ablation studies extended to 100 epochs, which validate the robustness of their design choices. The suggestion to improve the clarity of Figure 1 remains as a minor recommendation, but it does not affect the technical soundness of the work. The paper presents a novel and effective approach to 3D occupancy prediction with strong experimental results. Given the significance of the contribution and the thoroughness of the rebuttal, I maintain my rating of 4.

**Limitations:**

yes

**Paper Formatting Concerns:**

Not found.

**Quality:**

3

**Strengths And Weaknesses:**

Strengths:

This paper proposes a novel 3D occupancy prediction method called ODG, which introduces a dual Gaussian query architecture to separately model static and dynamic elements in the scene, demonstrating strong technical innovation. The method leverages a hierarchical Gaussian Transformer to perform coarse-to-fine predictions, effectively enhancing the model's representational capacity. Experimental results show that ODG achieves SOTA performance on both the Occ3D-nuScenes and Occ3D-Waymo datasets while maintaining low inference cost, highlighting the effectiveness and practicality of the approach.

Weaknesses:

1. The coarse-to-fine prediction mechanism in Figure 1 is not intuitively illustrated.
2. The inference speed of ODG-L is only 4.9 FPS, indicating that its superior performance comes at a significant cost of efficiency, which may hinder its deployment in real-world applications.
3. The ablation studies were conducted by training for only 24 epochs on Occ3D-nuScenes, while the final model was trained for 100 epochs; therefore, the validity of generalizing the conclusions remains uncertain.

---

> ### Author Rebuttal · Authors · 2025-07-30
>
> We thank reviewer JcRJ for the constructive feedbacks. We address the reviewer's questions and concerns below.
>
> $\textbf{Coarse-to-fine mechanism}$:
>
> We present our coarse-to-fine refinement scheme in **Algorithm 1** below. For notation simplicity, we denote motion compensation, feature sampling, cross and self attention, along with dual query-attention presented in Figure. 1 in the main paper as a single layer, rest of the notations are the same as defined in the main paper. To further enhance clarity, as described in the main paper, since coarse-to-fine refinement doesn't apply to bounding box attributes (i.e. $K_0=1$ throughout all the layers $\hat{\mathcal{T}}_l$), we have omitted them in **Algorithm 1** presented below.
>
> We note that due to OpenReview's limited TeX support, we had to declare transient notations such as $p,\text{inputs}^s_p,\text{ and }\text{inputs}_p^d$ to facilitate algorithmic flow.
>
> **Algorithm 1: Coarse-to-fine refinement in ODG**
>
> $\text{Input}: \text{image features } \textbf{F}; \text{static Guassian queries } \textbf{G}_0^s, \textbf{Q}_0^s; \text{dynamic Guassian queries } \textbf{G}_0^d, \textbf{Q}_0^d$
>
> $\text{Output}: \text{refined static Guassian queries } \textbf{G}_L^s, \textbf{Q}_L^s, \text{ class predictions }\mathbf{C}^s_L; \text{ refined dynamic Guassian queries } \textbf{G}_L^d, \textbf{Q}_L^d, \text{ class predictions }\mathbf{C}^d_L$
>
> $\text{Function } \text{ODGRefine}(\mathbf{F}; \textbf{G}_0^s, \textbf{Q}_0^s; \textbf{G}_0^d, \textbf{Q}_0^d):$
>
> $\quad\mathbf{G}^s_{:\mu,0}\in\mathbb{R}^{S\times K_0\times 3} \leftarrow \mathcal{U}(0, 1), K_0=1$
>
> $\quad\mathbf{G}^d_{:\mu,0}\in\mathbb{R}^{D\times K_0\times 3} \leftarrow \mathcal{U}(0, 1), K_0=1$
>
> $\quad$// initialize Gaussian means with uniform distribution; rest of the Gaussian parameters left uninitialized
>
> $\quad\text{for }l\leftarrow 1 \text{ to } L \text{ do }$
>
> $\quad\quad p=l-1$
>
> $\quad\quad\text{inputs}^s_p=[\mathbf{G}^s_{\:\mu,p}, \mathbf{Q}^s_{p}, \mathbf{F}]$
>
> $\quad\quad\mathbf{G}_{\:\mu, l}^s,\mathbf{Q}^s_l,\mathbf{C}^s_l=\hat{\mathcal{T}}_l(\text{inputs}^s_p)$
>
> $\quad\quad\mathbf{G}^s_{\mu:,l}=\Phi(\mathbf{G}^s_{:\mu,\ell}, \mathbf{Q}^s_l)$ // rest of Gaussian params predicted by MLP $\Phi$
>
> $\quad\quad\text{// }\mathbf{G}^s_{\:\mu, l}\in\mathbb{R}^{S\times K_l\times 3}, \mathbf{G}^s_{\mu\:, l}\in\mathbb{R}^{S\times K_l\times 10}, \mathbf{C}^s_l\in\mathbb{R}^{S\times K_l\times C}, K_p < K_l$, # coarse-to-fine
>
> $\quad\quad\text{inputs}^d_p=[\mathbf{G}^s_{\:\mu,p}, \mathbf{Q}^d_{p}, \mathbf{F}]$
>
> $\quad\quad\mathbf{G}_{\:\mu, l}^d, \mathbf{Q}_l^d,\mathbf{C}^d_l=\hat{\mathcal{T}}_l(\text{inputs}^d_p)$
>
> $\quad\quad\mathbf{G}^d_{\mu:,l}=\Phi(\mathbf{G}^d_{:\mu,\ell}, \mathbf{Q}^d_\ell)$ // rest of Gaussian params predicted by MLP $\Phi$
>
> $\quad\quad\text{// }\mathbf{G}^d_{\:\mu, l}\in\mathbb{R}^{D\times K_l\times 3}, \mathbf{G}^d_{\mu\:, l}\in\mathbb{R}^{D\times K_l\times 10}, \mathbf{C}^d_l\in\mathbb{R}^{D\times K_l\times C}, K_p < K_l$, # coarse-to-fine
>
> $\quad\text{return }[\mathbf{G}^s_l, \mathbf{Q}^s_l, \mathbf{C}^s_l], [\mathbf{G}^d_l, \mathbf{Q}^d_l, \mathbf{C}^d_l]$
>
> $\textbf{Efficiency analysis for ODG}$:
>
> We profiled both ODG-L and ODG-T using DeepSpeed. The results are summarized Table 1 and 2 below.
>
> **Table. 1 ODG-L profiling.**
> | component       |  runtime (ms)    |    percentage    |
> | -------- | ------- | -------- |
> |    img_backbone (ResNet-50)    |    33.82    |   16.58\%     |
> |    img_neck (FPN)    |    9.83    |   4.82\%     |
> |    ODG-L transformer  |    160.32   |   78.59\%     |
>
> **Table. 2 ODG-T profiling.**
> | component       |  runtime (ms)    |    percentage    |
> | -------- | ------- | -------- |
> |    img_backbone (ResNet-50)    |    12.30    |   24.88\%     |
> |    img_neck (FPN)    |    3.44    |   6.96\%     |
> |    ODG-T transformer    |    33.71  |   68.15\%     |
>
> We can see that in both ODG-L and ODG-T, the transformer part takes up most of the inference cost. We provide further profiling results on ODG-L transformer in Table 3 below. One can see that query self-attention takes up almost half the runtime. One straight-away optimization can be replacing self-attention with efficient attention schemes such as linear attention [1] or state space models (SSMs) [2], which would improve inference runtime. We will look into this as part of future work.
>
> **Table. 3 ODG-L transformer profiling.**
> | component       |  runtime (ms)    |    percentage    |
> | -------- | ------- | -------- |
> |    Self Attention    |     70.77   |   44.14\%     |
> |    Point Sampling    |   25.88   |   16.14%     |
> |    Cross Attention  |    24.06   |   15.01\%     |
> |     FFN                |         8.55         |        5.33\%           |
> |    ... |     ...     |       ...      |   ... |
>
> [1] Wang, Sinong, et al. "Linformer: Self-attention with linear complexity." arXiv preprint arXiv:2006.04768 (2020).
>
> [2] Gu, Albert, and Tri Dao. "Mamba: Linear-time sequence modeling with selective state spaces." arXiv preprint arXiv:2312.00752 (2023).
>
> $\textbf{Ablation studies for 100 epochs}$:
>
> We have now conducted all ablation experiments with ODG-T and trained for 100 epochs on nuScenes. The results are listed in Table 4 below. We can see under longer training time our ODG still consistently improves, which validates the efficacy of our designs. The conclusions on the effectiveness of our proposed components do not change when training for 24 or 100 epochs.
>
> **Table 4. Ablation studies of different components of ODG-T. Trained for 100 epochs on nuScenes.**
> | Motion compensation    | Query Attention | Rendering Sup | $\text{mIoU}$ | $\text{RayIoU}_{1m}$ | $\text{RayIoU}_{2m}$ | $\text{RayIoU}_{4m}$ | $\text{RayIoU}$ |
> | -------- | ------- | -------- | -------- | -------- | -------- | -------- | -------- |
> |   |     |  | 32.71 |30.0 | 39.6 | 42.7 | 37.4 |
> | yes |     |  | 33.97 | 31.1 | 40.3 | 43.5 | 38.5 |
> | yes  | yes    |  | 34.83 | 31.4 | 40.8 | 44.1 | 38.8
> | yes  | yes    | yes | 35.54 | 31.8 | 41.2 | 44.5 | 39.2 |

---

> > ### Comment · Reviewer_JcRJ · 2025-08-06
> >
> > Thank you for the detailed rebuttal. The additional algorithm description, efficiency analysis, and 100-epoch ablation study have addressed my concerns well, though I encourage the authors to further clarify Figure 1 for better visualization. I appreciate the clarity and rigor added to the paper. I maintain my rating.

---

> > > ### Author Response · Authors · 2025-08-08
> > >
> > > Dear Reviewer JcRJ,
> > >
> > > Thank you for recognizing our efforts in addressing your concerns! We truly appreciate your constructive feedback. Your insights were invaluable, not only for improving the manuscript but also for giving us the opportunity to reflect more deeply on our research.
> > >
> > > Best,
> > >
> > > Authors

---

### Official Review · Reviewer_cwc4 · 2025-07-03

**Clarity:** 3
**Significance:** 3
**Originality:** 3
**Rating:** 4
**Confidence:** 3

**Summary:**

This paper proposes the ODG framework for occupancy prediction. First, it uses priors of static and dynamic objects in the real world to initialize Gaussians for modeling. Second, the method designs a hierarchical model structure to improve modeling accuracy. Finally, multiple supervision signals are adopted during training. This approach enhances the accuracy of occupancy prediction without sacrificing inference speed.

**Questions:**

1. Regarding Weakness 1: Please elaborate on the design of the coarse-to-fine hierarchical structure.
2. Regarding Weakness 2: Please provide detailed visual comparisons with other methods.

Note: As I am not an expert in this field, I will also refer to suggestions from more specialized reviewers during the rebuttal stage.

**Ethical Concerns:**

["NO or VERY MINOR ethics concerns only"]

**Final Justification:**

The authors have provided a thorough response to our first question during the rebuttal period. However, due to changes in the conference policies, they were unable to address the second question. We are confident that the authors can supplement the relevant experimental details in subsequent stages, and thus we propose to maintain the "Borderline accept" rating.

**Limitations:**

Yes

**Paper Formatting Concerns:**

No formatting concern.

**Quality:**

3

**Strengths And Weaknesses:**

## Strengths
1. **Clear writing:** The paper is clearly written, allowing readers to easily follow the author’s logical flow.
2. **Superior performance:** The method outperforms state-of-the-art approaches across all listed test datasets.
3. **Good efficiency:** The proposed framework ensures competitive inference speed with minimal parameter overhead, balancing performance and efficiency.

## Weaknesses
1. **Model design details:** The authors highlight "Hierarchical Coarse-to-Fine Refinement" as a core contribution in the paper’s significance section, but the methodological description lacks detailed elaboration. Further clarification is expected.
2. **Visualization experiments:** The paper only provides visualization results of the proposed method. Visual comparisons between ODG and other methods are needed for a comprehensive evaluation.

---

> ### Author Rebuttal · Authors · 2025-07-30
>
> We thank reviewer cwc4 for the constructive feedbacks. We address the reviewer's questions and comments below.
>
> $\textbf{Design of coarse-to-fine hierarchical structure}$:
>
> We present our coarse-to-fine refinement scheme in **Algorithm 1** below.
> For notation simplicity, we denote motion compensation, feature sampling, cross and self attention, along with dual query-attention presented in Figure. 1 in the main paper as a single layer $\hat{\mathcal{T}}_l$, rest of the notations are the same as defined in the main paper. To further enhance clarity, as described in the main paper, since coarse-to-fine refinement doesn't apply to bounding box attributes (i.e. $K_0=1$ throughout all the layers $\hat{\mathcal{T}}_l$), we have omitted them in **Algorithm 1** presented below.
>
> We note that due to OpenReview's limited TeX support, we had to declare transient notations such as $p, \text{inputs}^s_{p}, \text{ and }\text{inputs}^d_{p}$ to facilitate algorithmic flow.
>
> **Algorithm 1: Coarse-to-fine refinement in ODG**
>
> $\text{Input}: \text{image features } \textbf{F}; \text{static Guassian queries } \textbf{G}_0^s, \textbf{Q}_0^s; \text{dynamic Guassian queries } \textbf{G}_0^d, \textbf{Q}_0^d$
>
> $\text{Output}: \text{refined static Guassian queries } \textbf{G}_L^s, \textbf{Q}_L^s, \text{ class predictions }\mathbf{C}^s_L; \text{ refined dynamic Guassian queries } \textbf{G}_L^d, \textbf{Q}_L^d, \text{ class predictions }\mathbf{C}^d_L$
>
> $\text{Function } \text{ODGRefine}(\mathbf{F}; \textbf{G}_0^s, \textbf{Q}_0^s; \textbf{G}_0^d, \textbf{Q}_0^d):$
>
> $\quad\mathbf{G}^s_{:\mu,0}\in\mathbb{R}^{S\times K_0\times 3} \leftarrow \mathcal{U}(0, 1), K_0=1$
>
> $\quad\mathbf{G}^d_{:\mu,0}\in\mathbb{R}^{D\times K_0\times 3} \leftarrow \mathcal{U}(0, 1), K_0=1$
>
> $\quad$// initialize Gaussian means with uniform distribution; rest of the Gaussian parameters left uninitialized
>
> $\quad\text{for }l\leftarrow 1 \text{ to } L \text{ do }$
>
> $\quad\quad p=l-1$
>
> $\quad\quad\text{inputs}^s_p=[\mathbf{G}^s_{\:\mu,p}, \mathbf{Q}^s_{p}, \mathbf{F}]$
>
> $\quad\quad\mathbf{G}_{\:\mu, l}^s,\mathbf{Q}^s_l,\mathbf{C}^s_l=\hat{\mathcal{T}}_l(\text{inputs}^s_p)$
>
> $\quad\quad\mathbf{G}^s_{\mu:,l}=\Phi(\mathbf{G}^s_{:\mu,\ell}, \mathbf{Q}^s_l)$ // rest of Gaussian params predicted by MLP $\Phi$
>
> $\quad\quad\text{// }\mathbf{G}^s_{\:\mu, l}\in\mathbb{R}^{S\times K_l\times 3}, \mathbf{G}^s_{\mu\:, l}\in\mathbb{R}^{S\times K_l\times 10}, \mathbf{C}^s_l\in\mathbb{R}^{S\times K_l\times C}, K_p < K_l$, # coarse-to-fine
>
> $\quad\quad\text{inputs}^d_p=[\mathbf{G}^s_{\:\mu,p}, \mathbf{Q}^d_{p}, \mathbf{F}]$
>
> $\quad\quad\mathbf{G}_{\:\mu, l}^d, \mathbf{Q}_l^d,\mathbf{C}^d_l=\hat{\mathcal{T}}_l(\text{inputs}^d_p)$
>
> $\quad\quad\mathbf{G}^d_{\mu:,l}=\Phi(\mathbf{G}^d_{:\mu,\ell}, \mathbf{Q}^d_\ell)$ // rest of Gaussian params predicted by MLP $\Phi$
>
> $\quad\quad\text{// }\mathbf{G}^d_{\:\mu, l}\in\mathbb{R}^{D\times K_l\times 3}, \mathbf{G}^d_{\mu\:, l}\in\mathbb{R}^{D\times K_l\times 10}, \mathbf{C}^d_l\in\mathbb{R}^{D\times K_l\times C}, K_p < K_l$, # coarse-to-fine
>
> $\quad\text{return }[\mathbf{G}^s_l, \mathbf{Q}^s_l, \mathbf{C}^s_l], [\mathbf{G}^d_l, \mathbf{Q}^d_l, \mathbf{C}^d_l]$
>
> $\textbf{Visual comparison with other methods}$:
>
> We have prepared such visual comparisons and will include them in the final version like in Fig. 2 and Fig. 3 to further demonstrate the superiority of our proposed ODG. Unfortunately, this year's NeurIPS rebuttal format does not support uploading a PDF for the rebuttal and as such, we are not able to share visualization results during the rebuttal stage.

---

> > ### Comment · Reviewer_cwc4 · 2025-08-02
> >
> > Thank you for your response. The authors have addressed our inquiries regarding the details of the model design. However, there have been adjustments to the submission policies concerning visualization this year, making it difficult for us to provide further evaluation. Therefore, we intend to maintain our original rating.

---

> > > ### Author Response · Authors · 2025-08-08
> > >
> > > Dear Reviewer cwc4,
> > >
> > > Thank you for recognizing our efforts in addressing your concerns! We truly appreciate your constructive feedback. Your insights were invaluable, not only for improving the manuscript but also for giving us the opportunity to reflect more deeply on our research.
> > >
> > > Best,
> > >
> > > Authors

---

### Comment · Area_Chair_Ms18 · 2025-08-04

Dear reviewers,

Please post your first response as soon as possible, so there is time for back and forth discussion with the authors.

All reviewers should respond to the authors, so that the authors know their rebuttal has been read.

Thanks,
AC

---

### Decision · Program_Chairs · 2025-09-17

**Decision:**

Accept (poster)

**Comment:**

This paper proposes a dual-query architecture comprising of two distinct sets of Gaussian queries to separately model the static and dynamic parts of the scene. A cross query attention is also introduced to establish interaction between queries, enhancing 3D occupancy prediction.

Strength:
Writing is good (cwc4, M77i), the performance is good (cwc4, JcRJ, M77i, XGZs), good balance between performance and computational complexity (cwc4, JcRJ, M77i), technical innovation is strong (JcRJ).

Weakness:
More implementation details are needed (cwc4, JcRJ), more detailed results are needed (XGZs). Novelty is not strong or ground-breaking (M77i, XGZs).

After rebuttal, this paper receives 4444. All the reviewers are satisfied with authors' response and give positive feedback. The AC agrees with reviewers, although not ground-breaking, this paper makes a relevant contribution to the field with its own novel designs. Please include the additional results and modifications here into later versions.